# MODEL VALIDATION USING MUTATED TRAINING LABELS: AN EXPLORATORY STUDY

## ABSTRACT

We introduce an exploratory study on Mutation Validation (MV), a model validation method using mutated training labels for supervised learning. MV mutates training data labels, retrains the model against the mutated data, then uses the metamorphic relation that captures the consequent training performance changes to assess model fit. It does not use a validation set or test set. The intuition underpinning MV is that overfitting models tend to fit noise in the training data. We explore 8 different learning algorithms, 18 datasets, and 5 types of hyperparameter tuning tasks. Our results demonstrate that MV is accurate in model selection: the model recommendation hit rate is 92% for MV and less than 60% for out-of-sample-validation. MV also provides more stable hyperparameter tuning results than out-of-sample-validation across different runs.

## 1 INTRODUCTION

Out-of-sample validation (such as test accuracy) is arguably the most popular approach adopted by researchers and developers for empirically validating models in applied machine learning. It uses data different from the training data to approximate future unseen data. However, out-of-sample validation is widely acknowledged to have limitations: 1) the sample set may be too small to represent the data distribution; 2) the accuracy can have a large variance across different runs (Pham et al., 2020); 3) the samples are typically randomly selected from the collected data, and may therefore have similar bias as in the training data, leading to an inflated validation score (Piironen & Vehtari, 2017; Gronau & Wagenmakers, 2019); 4) excessive reuse of a fixed set of samples can lead to overfitting even if the samples are held out and not used in the training process (Feldman et al., 2019).

The Mutation Validation (MV) approach we explore is a new approach to validating machine learning models relying *only* upon training data. MV applies Mutation Testing and Metamorphic Testing —two software engineering techniques that validate code correctness. Mutation testing mutates the program by making synthetic changes (e.g., to change $a + b$ into $a - b$ or to remove a functional call), then re-executes the tests to monitor the behaviour changes and check the power of a test suite (Papadakis et al., 2019; Jia & Harman, 2010). Metamorphic Testing detects program errors by checking metamorphic relations, which is the relationship between input changes and output changes (Chen et al., 2020; Segura et al., 2016). Combining these two techniques, MV mutates training data labels and retrains the model using the mutated data, then measures the training performance change. As shown by Figure 1, the key intuition is that a learner, if fitting the given training data property, would be less likely to be 'fooled' by a small ratio of mutated labels. Consequently, the model trained with the mutated training data would "detect" the mutated labels and still exhibit high predictive performance on the original training data. By contrast, an overfitted learner violates the Occam's Razor (Hawkins, 2004), and has extra capacity to fit incorrect noisy labels, and thus will yield a model that exhibits poor predictive performance on the original training data, but high performance on the mutated training data. Furthermore, an over-simple learner has poor learnability, and will have low performance on the original training data and the mutated data.

There are a significant number of theories proposed to explain model validation and complexity, such as Rademacher complexity and VC dimension. Nevertheless, the prescriptive and descriptive value of these theories remains debated. Zhang et al. (2017) found that deep hypothesis spaces can be large enough to memorise random labels. They discussed the limitations of existing measurements in explaining the generalisation ability of large neural networks, and called for new measurements.

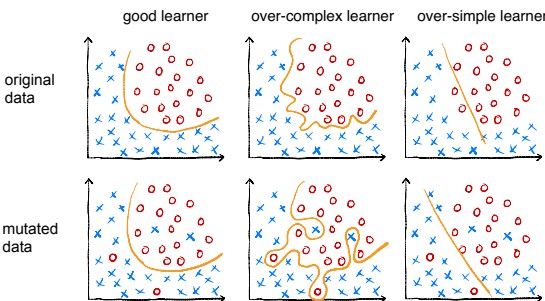

Figure 1: The intuition underpinning MV: A better learner is less affected by the mutated labels.

In this present work, we do not compare MV with these theories. Rather, similar to Zhang et al. (2017), we focus on empirical investigation. In Appendix A.1, we provide the theoretical foundations underpinning the metamorphic relation that MV uses.

We report on the performance of MV on 12 open datasets and 6 synthetic datasets with different known data distributions (see Table 1), using 8 widely-adopted classifiers (including both classic learning classifiers and deep learning classifiers). We investigate the effectiveness and stability of MV serving as a complementary measure to the existing practical model validation methods for model selection and parameter tuning. The experimental results provide evidence to support the following conclusions. **First**, MV captures well the degree of match between decision boundaries and data patterns in model selection. The model recommendation hit rate for MV is 92%, but is 53% for cross validation, and 55% for test accuracy. **Second**, MV is more responsive to changes in capacity than conventional validation methods. When cross validation (CV) accuracy and test accuracy could not distinguish among large-capacity hyperparameters, MV complements them in hyperparameter tuning. **Third**, MV is stable in model validation results. Its dropout rate tuning result does not change across five runs; the variance is zero. For validation set and test set, the average variance is 0.003 and 0.007, respectively. The paper also discusses the connections between MV and other noise injection work in the literature, as well as the usage scenarios of MV (Section 5).

In summary, we make the following primary contributions:
**1) An exploratory study on the performance of Mutation Validation.** We explore the effectiveness and stability of MV to validate machine learning (ML) models as a complement to the currently used empirical model validation methods. MV requires neither validation nor test sets, but uses the training performance sensitivity to the mutated training labels. We study 18 datasets, 8 different learning algorithms, and 5 hyperparameter tuning tasks.
**2) An application of software testing techniques in ML model validation.** MV is the first approach that applies mutation testing and metamorphic testing —two widely studied software testing techniques —on model validation tasks.

## 2 MUTATION VALIDATION

### 2.1 GENERAL INTUITION

Figure 1 illustrates the intuition that underpins MV. For a 'good' learner that fits the training data well (first column of Figure 1), the learner is less likely to be 'fooled' by a small number of mutated labels, and would keep predicting the original labels for the mutated samples. As a result, the model trained on the mutated data will still have high predictive accuracy on the original training data, but will have decreased predictive accuracy on the mutated data. An overfitted learner tends to fit noise in the training data (second column of Figure 1). With mutated data, the learner will yield a model that makes predictions following the incorrect labels, leading to a high training accuracy on the mutated data, but poor accuracy on the original data. An over-simple learner has poor learnability. The model it yields has poor performance with or without the mutated training data. As a result, the model trained on the mutated data has low accuracy with both the original labels and the mutated labels.

## 2.2 MUTATION VALIDATION

MV uses mutation testing and metamorphic testing, two software validation techniques, to validate ML models. Mutation testing creates mutants by injecting faults in a program, then re-executes the program to check whether a test suite detects those faults. Metamorphic relation specifies how a change in the input should result in a change in the output. It is used to detect errors in software when there are no reliable oracles. For a program $f$, its input $x$ and $x'$. Let $f(x)$ and $f(x')$ be the execution outputs of $x$ and $x'$ against $f$. Let $\mathbb{R}_i$ be the relationship between $x$ and $x'$, $\mathbb{R}_o$ is the relationship between $f(x)$ and $f(x')$. A metamorphic relation can be represented as: $\mathbb{R}_i(x, x') \Rightarrow \mathbb{R}_o(f(x), f(x'))$. If the relation is violated, the program under test contains a bug. For example, when validating the $\sin$ mathematical function, one metamorphic relation that a correct program should hold is $\sin(x + \pi) = -\sin(x)$.

Now consider the scenario of ML model validation. If we treat a learner as the program under test, the training data as the input, the trained model's behaviours as the output, our intuition introduced in Section 2.1 can be well captured by mutation testing and metamorphic relations. In particular, the input changes are introduced by mutating training data, each mutated data instance is called a mutant; the output changes are defined in terms of performance changes of the trained models. Based on this, we propose **Mutation Validation**, a new machine learning model validation method that validates ML models based on the relationship between training data changes and training performance changes. *A good learner is expected to "detect" the mutants and have a certain amount of training performance changes according to the number of input data changes.*

There are different metamorphic relations that can be explored to conduct MV, which may depend on the data mutation method, the training performance measurement (e.g., accuracy, precision, loss), and the calculation of performance changes. Let $\eta$ be the mutation degree (i.e., the ratio of randomly mutated labels in the training data). $S$ is the original training data, $S_\eta$ is the mutated training data with mutation degree $\eta$ ($\eta \leq 0.5$), $f(S)$ is the model output trained on $S$, $f(S_\eta)$ is the model output trained on $S_\eta$, $\widehat{T}_S(f(S))$, $\widehat{T}_S(f(S_\eta))$ are the accuracy of $f(S)$ and $f(S_\eta)$ based on the original training labels, respectively. $\widehat{T}_{S_\eta}(fS_{(\eta)})$ is the accuracy of $f(S_\eta)$ based on the mutated labels. In this present work, as the first exploratory study on MV, we study the following MV measurement $m$ to conduct model validation:

$$m = (1 - 2\eta)\widehat{T}_S(f(S_\eta)) + \widehat{T}_S(f(S)) - \widehat{T}_{S_\eta}(f(S_\eta)) + \eta. \tag{1}$$

The above measurement, although derived from theory (Appendix A.1), matches well with our intuition introduced in Section 2.1. In particular, if the learner is less affected by the mutated labels, the predictive behaviours of the trained model with mutated labels should be closer to that of the model trained with the original labels. This leads to a larger $\widehat{T}_S(f(S_\eta))$ and a larger difference between $\widehat{T}_S(f(S))$ and $\widehat{T}_{S_\eta}(f(S_\eta))$, as long as the mutation degree $\eta$ is fixed. The larger $m$ is, the better the learner fits the training data. In the optimal case, the trained model on mutated data has a perfect training accuracy on the original training labels (i.e., $\widehat{T}_S(f(S_\eta)) = 1$), and detects all the mutants (i.e, $\widehat{T}_S(f(S)) - \widehat{T}_{S_\eta}(f(S_\eta)) = \eta$). Thus, $m = 1$. Mutation degree $\eta$ ranges between 0 and 0.5, but needs to be a fixed value. The theory inspiration of this metric, its metamorphic relation, as well as the influence of $\eta$ on $m$ are provided in our appendix.

## 3 EXPERIMENTAL SETUP

The main body of this paper answers four research questions:

*RQ1: What is the effectiveness of MV in model selection?*
*RQ2: What is the effectiveness of MV in hyperparameter tuning?*
*RQ3: What is the stability of MV in validating machine learning models?*
*RQ4: How does training data size affect MV?*

We also explore the efficiency of MV in validating machine learning models, as well as the influence of mutation degree $\eta$ on MV. The details are in the appendix.

To evaluate MV, we choose to use datasets that are diverse in category, size, class number, feature number, and class balance situations. Small datasets are particularly important to demonstrate the

| Dataset | abbr. | #training | #test | #class | #feature |
|---|---|---|---|---|---|
| synthetic-moon | moon | $100 - 1e+6$ | 2,000 | 2 | 2 |
| synthetic-moon (0.2 noise) | moon-0.2 | $100 - 1e+6$ | 2,000 | 2 | 2 |
| synthetic-circle | circle | $100 - 1e+6$ | 2,000 | 2 | 2 |
| synthetic-circle (0.2 noise) | circle-0.2 | $100 - 1e+6$ | 2,000 | 2 | 2 |
| synthetic-linear | linear | $100 - 1e+6$ | 2,000 | 2 | 2 |
| synthetic-linear (0.2 noise) | linear-0.2 | $100 - 1e+6$ | 2,000 | 2 | 2 |
| Iris | iris | 150 | – | 3 | 4 |
| Wine | wine | 178 | – | 3 | 13 |
| Breast Cancer Wisconsin | cancer | 569 | – | 2 | 9 |
| Car Evaluation | car | 1,728 | – | 4 | 6 |
| Heart Disease | heart | 303 | – | 5 | 14 |
| Bank Marketing | bank | 45,211 | – | 2 | 17 |
| Adult | adult | 48,842 | 16,281 | 2 | 14 |
| Connect-4 | connect | 67,557 | – | 2 | 42 |
| MNIST | mnist | 60,000 | 10,000 | 10 | – |
| fashion MNIST | fashion | 60,000 | 10,000 | 10 | – |
| CIFAR-10 | cifar10 | 50,000 | 10,000 | 10 | – |
| CIFAR-100 | cifar100 | 50,000 | 10,000 | 100 | – |

Table 1: Details of the datasets.

ability of MV in providing warnings for over-complex learners. Table 1 shows the details of each dataset used to evaluate MV. Column "#training" and Column "#test" show the size of training data and test data, which is presented by the dataset providers. Column "#class" shows the number of classes (or labels) for each dataset. Column "#feature" presents the number of features.

To obtain datasets with known **ground-truth** decision boundaries for validating model fitting, we use synthetic datasets with three types of data distributions: moon, circle, and linearly-separable. These three data distributions are provided by *scikit-learn* (Pedregosa et al., 2011) tutorial to demonstrate the decision boundaries of different classifiers. To study the influence of original noise in training data on MV, for each type of distribution, we create datasets with noise. We also generate different-size training data ranging from 100 to 1 million data points to study the influence of training data size. These synthetic datasets help check whether MV identifies the right model whose decision boundary matches the data distribution, with and without noise in the original dataset $S$. We do not expect such synthetic datasets to reflect real-world data, but the degree of control and interpretability they offer allows us to verify the behaviour of MV with a known ground-truth for model selection.

We also report results on 12 real-world widely-adopted datasets with different sizes, numbers of features and classes. Eight of them are from the UCI repository (Asuncion & Newman, 2007), the remaining four are the most widely used image datasets: MNIST, fashionMNIST, CIFAR-10, and CIFAR-100. For each dataset, we calculate MV with $\eta = 0.2$ to answer the research questions. That is, under each class of the dataset, we randomly select 20% of the labels to mutate. This guarantees that MV is not affected by the problem of class imbalance. We mutate the labels by label swapping: to replace a label with the next label in the label list, the final label in the label list is replaced with the first label in the label list. For the synthetic datasets and the 8 UCI datasets, we compare with 3-fold CV accuracy and test accuracy. We use 3-fold CV because it has almost identical results to more folds cross validation, but has lower cost, thereby being more conservative to compare with when studying the efficiency of MV. For the three image datasets on deep learning models, we compare with validation accuracy with 20% validation data (split out from the training data) and test accuracy. The next section introduces more configuration details.

## 4 RESULTS

### 4.1 RQ1: EFFECTIVENESS OF MV IN MODEL SELECTION

To answer RQ1, we use synthetic datasets to explore whether MV recommends the learners whose decision boundaries best match the data patterns. We use synthetic datasets because synthetic datasets have ground-truth data patterns for model selection (more details in Section 3). The data distribution of real-world UCI datasets, however, is often unknown or difficult to visualise. To generate synthetic datasets, we use three *scikit-learn* (Pedregosa et al., 2011) synthetic dataset distributions, which are designed as a tutorial to illustrate the nature of decision boundaries of different classifiers. This experiment uses the same settings as in the *Scikit-learn* tutorial (Classifier Comparison, 2019). Each dataset has 100 training data points for model selection. We generate another 2,000 points as test sets.

Figure 2 shows the training data points, decision boundaries of each classifier, and measurement values from MV, CV, and test accuracy (on the 2,000 test data points). We have the following

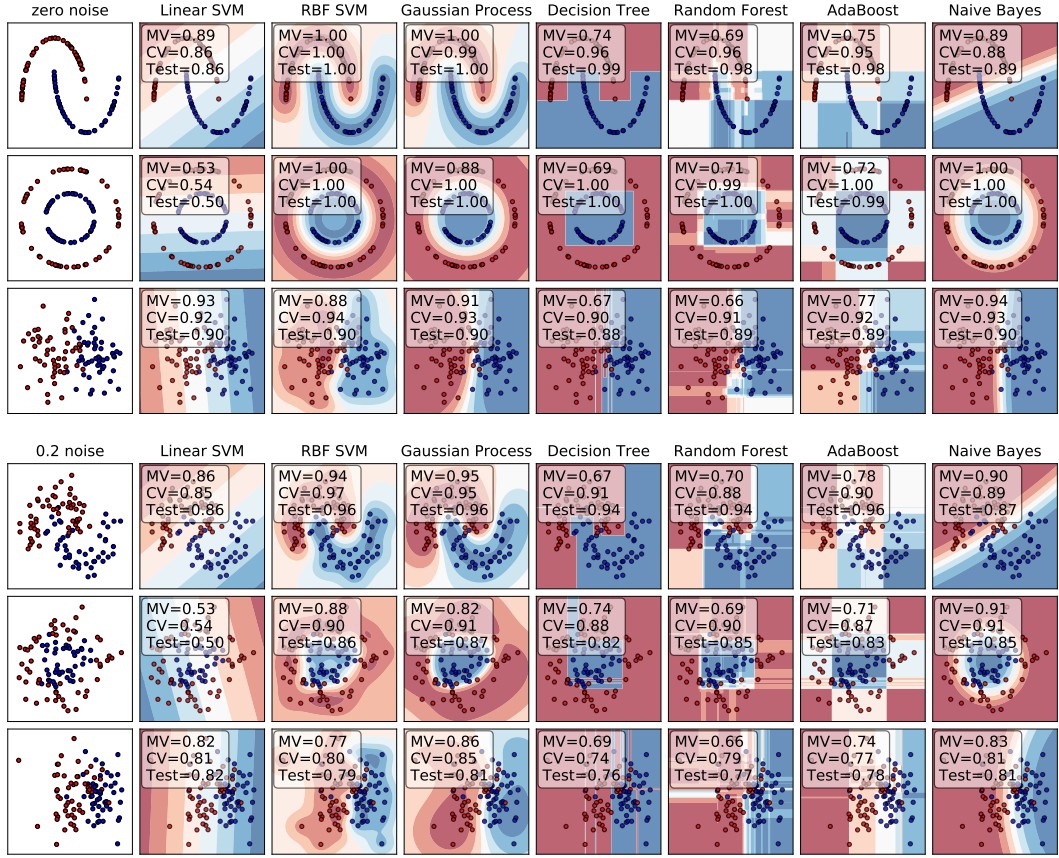

Figure 2: Performance of MV in model selection. MV, CV, Test denote MV score, CV accuracy, and Test Accuracy. Red and blue points are the original training data without noise (top-three rows) and with 0.2 noise (bottom-three rows). Areas with different colours show the decision boundaries. We observe that **MV captures well the match between decision boundaries and data patterns**.

observations. *First*, MV tends to have large values for cases where the decision boundaries match well the data patterns; it provides more discriminating scores for model selection under different datasets, it is easy to pick out the top-2 models that match the data distributions the best based on MV. *Second*, the recommended models from MV are less affected by the noise in the training data.

The figure also shows cases *where CV and test accuracy have limitations* in evaluating ML models. For example, in Figure 2, for the circle distribution, Decision Trees and Random Forests (with maximum depth of 10) give obviously ill-fitted rectangle-shaped decision boundaries, yet the cross validation accuracy and test accuracy remain high. In addition, from Table 2, it is more difficult to select proper models based on CV and test accuracy, because CV and test accuracy are often very similar across different models.

For ease of observation, we present Table 2 to show the models recommended by MV, CV, and test accuracy, based on their top-2 scores shown by Figure 2. The ground-truth models in bold are based on manual observation and widely-adopted ML knowledge. For example, *Scikit-learn* documentation (Classifier Comparison,

| Dataset | Method | Noise | Recommended models based on top-2 scores |
|---|---|---|---|
| moon | MV | 0.0 | **RBF SVM**, **Gaussian Process** |
| | | 0.2 | **RBF SVM**, **Gaussian Process** |
| | CV | 0.0 | **RBF SVM**, **Gaussian Process** |
| | | 0.2 | **RBF SVM**, **Gaussian Process** |
| | Test | 0.0 | **RBF SVM**, **Gaussian Process** |
| | | 0.2 | **RBF SVM**, **Gaussian Process**, AdaBoost |
| circle | MV | 0.0 | **RBF SVM**, **Naive Bayes** |
| | | 0.2 | **RBF SVM**, **Naive Bayes** |
| | CV | 0.0 | **RBF SVM**, Gaussian Process, DT, RF, AdaBoost, **Naive Bayes** |
| | | 0.2 | Gaussian Process, **Naive Bayes** |
| | Test | 0.0 | **RBF SVM**, Gaussian Process, DT, RF, **Naive Bayes** |
| | | 0.2 | **RBF SVM**, Gaussian Process |
| linear | MV | 0.0 | **Linear SVM**, **Naive Bayes** |
| | | 0.2 | Gaussian Process, **Naive Bayes** |
| | CV | 0.0 | RBF SVM, Gaussian Process, **Naive Bayes** |
| | | 0.2 | Gaussian Process, **Naive Bayes** |
| | Test | 0.0 | **Linear SVM**, RBF SVM, Gaussian Process, RF, **Naive Bayes** |
| | | 0.2 | **Linear SVM**, Gaussian Process, **Naive Bayes** |

Table 2: Recommended models by MV, CV, and test accuracy. The models in bold are the ground-truth ones whose decision boundaries match the data patterns. The recommendation hit rate is 92% for MV, 53% for CV, 55% for test accuracy.

2019) mentions that Naive Bayes and Linear SVM are more suitable for linearly-separable data. Overall, we have the following conclusion: among all the model selection tasks we explored, the hit rate (i.e., the ratio of recommended models that match the ground truth model) is 92% for MV, 53% for CV accuracy, and 55% for test accuracy.

## 4.2 RQ2: EFFECTIVENESS OF MV IN HYPERPARAMETER CONFIGURATION

For models whose capacity increases along with their hyperparameters, we expect their goodness of model fit to first increase, then peak, before decreasing. The peak indicates the best model fit assessed according to MV. In RQ2, we assess whether MV matches this pattern. We study five capacity-related hyperparameters for several widely-adopted algorithms: the maximum depth for a Decision Tree, C and gamma for a Support Vector Machine (SVM), and the dropout rate and learning rate for Convolutional Neural Networks (CNNs).

**Maximum Depth for Decision Tree.** Figure 3 shows how MV responds to increases in maximum depth of Decision Tree for the eight real-world UCI datasets. For small datasets (smaller than 2,000), we do not split out test data, but take the whole data as training data. For the bank and connect datasets, we use 80% of the data as test sets. For adult, we use its original test set. We repeat the experiments 10 times. The yellow shadow in the figure indicates the variance across the 10 runs.

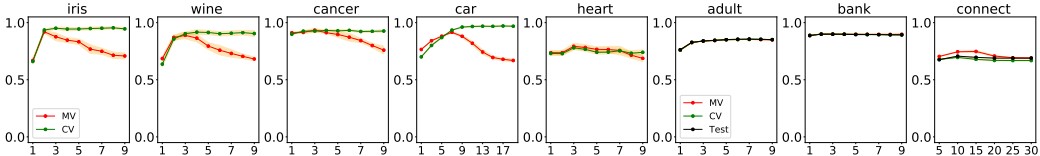

Figure 3: Changes in MV, CV accuracy, and test accuracy when increasing the *maximum depth* for Decision Trees. The x-axis ticks for car and connect differ to capture MV's inflection point. We can observe that, while CV and test accuracy agree with MV on the key influence point in most cases, they are less responsive to large depths (which lead to overfitting).

From Figure 3, we make the following observations. First, for 6 out of the 8 datasets, MV increases then decreases as maximum depth increases, and exhibits a maximum in each curve. This is consistent with the pattern we expect a good measure to exhibit. For small datasets, large depths yield low MV scores, whereas large datasets do not. For adult and bank, MV values remain large when the maximum depth increases. We suspect that this is because, for these two datasets, the training data size is large enough for the model to obtain resilience against mutated labels. We explore the influence of training data size in the appendix.

We also observe that MV, CV and test accuracy have similar inflection points, yet MV is more responsive to depth changes, especially when the model overfits due to large depths. This observation indicates that MV provides further information to help tune maximum depth when CV and test accuracy are less able to distinguish multiple parameters. In particular, if a developer uses grid search to select the best maximum-depth ranged between 5 and 10, we find that grid search suggests depths of 8, 6, 9 in three runs for the cancer dataset, which are over-complex and unstable. Similar results are observed for other small datasets. With MV, its decrease trend in this range indicates that there is a simpler model with comparable predictive accuracy but better resilience to label mutation.

**C and gamma for SVM.** In SVM, the gamma parameter defines how far the influence of a single training example reaches; the C parameter decides the size of the decision boundary margin, behaving as a regularisation parameter. In a heat map of MV scores as a function of C and gamma, the expectation is that good models should be found close to the diagonal of C and gamma (Scikit-learn:SVM, 2020). Figure 4 presents the heat map for cross validation and MV for two datasets. We do not use a hold out test set to ensure sufficient training data. The upper left triangle in each sub-figure denotes small complexity; the bottom right triangle in each sub-figure denotes large complexity. In both cases, MV gives low scores.

When comparing CV and MV, MV is more responsive to hyperparameter value changes. With MV scores, it is more obvious that good models can be found along the diagonal of C and gamma. When C and gamma are both large, the CV score is high but MV score is low, this is an indication that there exists a simpler model with similar test accuracy. In practice, as stated by Scikit-learn documentation,

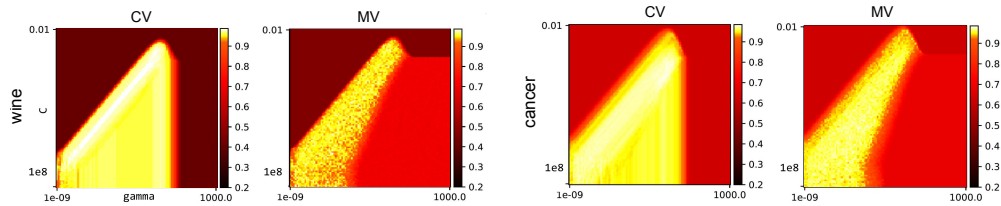

Figure 4: Influence of SVM parameters on CV and MV. The horizontal/vertical axis is gamma/C. Good models are expected to be found close to the diagonal. As can be seen, CV has a broad high-valued (bright) region, while MV's high-valued region (bright) is narrower, showing that MV is more responsive to parameter changes.

it is interesting to simplify the decision function with a lower value of C so as to favour models that use less memory and that are faster to predict (Scikit-learn:SVM, 2020).

**Dropout Rate and Learning Rate for CNN.** We use CNN models coming from Keras documentation. Validation accuracy is calculated with 80% training data and 20% validation data. Figure 5 shows the results. We observe that when tuning dropout rate for mnist and fashion-mnist, validation and test accuracy are less discriminating and provide different tuning results across different runs (more details in Table 4), yet MV is more discriminating. For dropout rate, MV and test accuracy have different key influence points on cifar10 (0.4 v.s. 0.2). This is because there is a big capacity jump between 0.2 and 0.4. The result suggests that the optimal dropout rate with comparable validation/test accuracy is between 0.2 and 0.4.

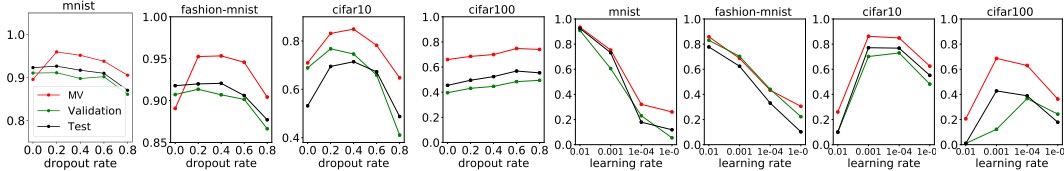

Figure 5: Influence of CNN's dropout rate and learning rate on MV and validation/test accuracy.

Overall, we observe that MV is responsive to hyperparameter changes on all the hyperparameter tuning tasks we explored. CV and test accuracy are less responsive especially for large-capacity hyperparameters (i.e., large maximum depth for Decision Tree and small dropout rate for CNNs). This leads to the following issues. First, due to the large variance of their results across different runs, such low response often leads to unstable hyperparameter recommendation results. We further explore this in Section 4.3. Second, developers may easily choose an over-complex learner, making the learner: 1) easily biased by incorrect training labels; 2) vulnerable to training data attack; 3) computationally and memory intensive; 4) more difficult to interpret.

### 4.3 RQ3: STABILITY OF MV IN MODEL VALIDATION

For the machine learning experimental design in the literature, data is usually random split. The values of MV, CV, and test accuracy may be easily affected by the randomness in model building (Pham et al., 2020), especially when the overall data set is small, or building deep learning models. As a result, with different runs, developers may get completely different recommendations for the best hyperparameter configuration. A good model validation method is expected to be stable in hyperparameter recommendation results, giving developers clear instructions on the best choice.

To explore the stability of MV, CV, and test accuracy, we run the model building process multiple times, each time with a different split of training data and validation data (the size of training/validation data is unchanged). We then record the recommended best hyperparameters during each run under the scenario of hyperparameter configuration. We use the maximum depth for Decision Tree with the 5 small UCI datasets as well as the dropout rate for CNN with the 3 image datasets. Table 4 shows the results, which lead to the following conclusion: MV has good stability in recommending hyperparameters. When recommending maximum depth for Decision Tree, the overall variance is

| Dataset | Method | Recommended maximum depth | Variance |
|---|---|---|---|
| iris | MV | [3, 3, 3, 2, 2, 2, 2, 3, 3, 2] | 0 |
| | CV | [7, 6, 6, 3, 3, 9, 3, 7, 5, 8] | 4 |
| wine | MV | [3, 3, 2, 3, 2, 3, 4, 3, 2, 2] | 0 |
| | CV | [3, 4, 7, 8, 3, 3, 4, 3, 5, 3] | 3 |
| cancer | MV | [3, 3, 3, 2, 3, 3, 3, 3, 3, 3] | 0 |
| | CV | [5, 7, 3, 4, 4, 4, 7, 4, 3, 6] | 2 |
| car | MV | [7, 7, 7, 7, 7, 7, 7, 7, 7, 7] | 0 |
| | CV | [11,19,19,15,17,13,11,13,13,15] | 8 |
| heart | MV | [5, 6, 3, 3, 6, 5, 6, 4, 3, 5] | 1 |
| | CV | [4, 4, 3, 3, 4, 4, 3, 3, 9, 5] | 3 |
| **mean** | **MV** | – | **0.200** |
| | CV | – | **4.000** |

| Dataset | Method | Recommended dropout rate | Variance |
|---|---|---|---|
| mnist | MV | [0.2, 0.2, 0.2, 0.2, 0.2, 0.2, 0.2, 0.2, 0.2, 0.2] | 0.000 |
| | Vali. | [0.0, 0.0, 0.2, 0.2, 0.0, 0.0, 0.2, 0.0, 0.0, 0.2] | 0.011 |
| | Test | [0.2, 0.0, 0.0, 0.2, 0.0, 0.0, 0.0, 0.0, 0.2, 0.0] | 0.009 |
| f-mnist | MV | [0.2, 0.2, 0.2, 0.2, 0.2, 0.2, 0.2, 0.2, 0.2, 0.2] | 0.000 |
| | Vali. | [0.2, 0.2, 0.2, 0.2, 0.2, 0.2, 0.2, 0.2, 0.2, 0.2] | 0.000 |
| | Test | [0.2, 0.0, 0.2, 0.0, 0.0, 0.0, 0.0, 0.2, 0.2, 0.0] | 0.011 |
| cifar10 | MV | [0.4, 0.4, 0.4, 0.4, 0.4, 0.4, 0.4, 0.4, 0.4, 0.4] | 0.000 |
| | Vali. | [0.2, 0.2, 0.2, 0.2, 0.2, 0.2, 0.2, 0.2, 0.2, 0.2] | 0.000 |
| | Test | [0.2, 0.2, 0.2, 0.2, 0.2, 0.2, 0.2, 0.2, 0.2, 0.2] | 0.000 |
| **mean** | **MV** | – | **0.000** |
| | Vali. | – | **0.003** |
| | Test | – | **0.007** |

Table 4: Recommended hyperparameters across different runs. The third column shows the specific recommended hyperparameters (we run the datasets 10 times); the last column shows the variance (the average of the squared differences from the Mean) across different runs. MV is more stable than CV and test accuracy in recommending hyperparameters.

0.200 for MV, but 4.000 for CV accuracy; when recommending dropout rate for CNN, the overall variance is 0.000 for MV, 0.003 for validation accuracy, and 0.007 for test accuracy.

### 4.4 RQ4: INFLUENCE OF TRAINING DATA SIZE ON MV

We expect that when a learner is over-complex for the data, adding extra data will improve the learner's resilience to mutated labels, thus the MV score ought also to increase when training data size increases. Indeed, previously from Figure 3, we have observed that models trained on large datasets (e.g., bank and adult) tend to be more resilient to mutated labels for large depths. Figure 6 shows the results for connect and moon datasets, indicating that data size plays a role in determining the trained model's robustness to incorrect training labels.

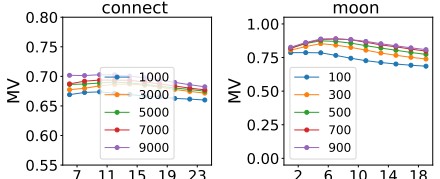

Figure 6: Influence of training data size on MV when increasing maximum depths (horizontal axis) of Decision Trees. For a given depth, larger datasets tend to yield larger MV values.

These observations indicate another possible value in the use of MV, as a complement to CV and test accuracy. Specifically, where MV is low, the ML engineers have two potential actions to improve the fitting between the learner and the training data: either to optimise the learner (e.g., search for smaller-capacity models with comparable test accuracy), or to optimise the data (e.g., increase data size to increase the learner's resilience to incorrect training labels).

## 5 DISCUSSION

### 5.1 CONNECTION WITH RELATED WORK

MV is also connected with several key concepts in the literature. This section discusses the connections as well as the differences between MV and these concepts.

**Noise injection in training data** has long been recognised as an approach to complex training data and reduce overfitting (Holmstrom et al., 1992; Greff et al., 2016). There are three main differences between MV and this random noise injection. 1) Overfitting prevention noise is often Gaussian noise (Greff et al., 2016), and is added into the training inputs (not labels); on the contrary, MV does not use random noise, but uses label swapping, the mutation is applied only on labels. 2) Conventional noise injection changes the training data, then directly uses the model trained on this noisy training data. MV keeps the original training data, but creates another mutated training data to for measurement calculation. This mutated training data will not be used to yield any model for real prediction tasks. 3) Conventional noise injection aims to improve the complexity of the training data, to reduce overfitting. MV aims to calculate a model validation measurement score, to measure

the goodness of model fitting. Zhang et al. (2017) adds noise in training data labels to study the generalisation of DNN, while MV mutates labels to provide model validation measurement.

**Noise injection in test data** is adopted to evaluate the robustness of a model. For example, the generation of adversarial examples (Goodfellow et al., 2014) uses noise injection in the test inputs. Compared to this technique, MV 1) mutates training data, not test data; 2) mutates labels, not features; 3) aims to validate model fitting, rather than model robustness to feature perturbations.

**Overfitting prevention** refers to the techniques adopted in the training process to avoid overfitting, especially when training deep neural networks. The key techniques are regularisation, early stopping, ensembling, dropout, and so on. However, as shown in this paper, conventional overfitting detection techniques, such as CV or validation accuracy, are often less responsive to overfitting. In practice, developers often conduct the *prevention* without knowing whether the overfitting happens or not. MV is demonstrated to be discriminating and stable in detecting over-complex learners. It thus provides signals for the adoption and configuration of these overfitting prevention techniques.

**Rademacher complexity**. In statistical machine learning, Rademacher complexity has been used to measure the complexity of a learner's hypothesis space. It also mutates training data. It measures how well the learner correlates with randomly generated labels on the training data, but can be difficult to compute (Rosenberg & Bartlett, 2007). Different from Rademacher complexity, MV is an applied tool, not a theoretical tool. MV uses label mutations for a part of the data. Furthermore, Rademacher complexity cares only about the training accuracy on the mutated data, but MV uses the accuracy changes on the original and the mutated data.

## 5.2 The Usage of MV in Practice

With our exploratory study, we find that MV is capable of complementing existing model validation techniques. There are two main application scenarios of MV: 1) When out-of-sample validation results are similar or unstable across different models or hyperparameters, MV can help to guide the selection process (see more in Section 4.2 and Section 4.3). 2) When the training data size is limited, MV is a better option than validation accuracy because it does not need to split out a validation set, thus reserving more data for training.

The test accuracy, although having limitations shown in the literature and our experiments, is still a very important method for developers to learn the generalisation ability of a model, especially when there is data distribution shift between the training data and unseen data. However, as shown in this paper, when test accuracy is high but MV is low, this means that there is a learner with comparable test accuracy, but is simpler.

MV deserves more attention from developers when simplicity, security (e.g., defending training data attack), and interpretibility of the built models are required. Theoretically, MV can be applied in other ML tasks which rely on out-of-sample validation, such as feature selection. We will explore the effectiveness of MV in these tasks in future work.

## 6 Conclusion

We introduced an exploratory study on MV, a new approach to assessing how good a learner fits the given training data. MV validates via checking the learner's sensitivity to training labels changes (expressed as label mutants). The sensitivity is captured by metamorphic relations. We show that MV is more effective and stable than the currently adopted CV, validation accuracy, and test accuracy. It is also responsive to model capacity and training data characteristics. These results provide evidence that MV complements the existing model validation practises. We hope the present paper will serve as a starting point to future contributions.

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

# A  APPENDIX

## A.1  THEORETICAL INSPIRATION

This part introduces the theory that leads to a metamorphic relation and formula that we use for conducting MV. The denotations and terms we used are guided by the standard statistical machine learning literature (Mohri et al., 2018; Ghosh et al., 2017).

**Theoretical Metamorphic Relation for Model Validation:**

Let $\mathcal{X}$ be the feature space from which the data are drawn. Let $\mathcal{Y} = [k] = \{1, ..., k\}$ be the class labels. Given a training set $S = \{(\mathbf{x_1}, y_{\mathbf{x_1}}), ..., (\mathbf{x_m}, y_{\mathbf{x_m}})\} \in (\mathcal{X} \times \mathcal{Y})^m$ which is independently and identically distributed according to some fixed, but unknown, distribution $\mathcal{D}$ of $\mathcal{X} \times \mathcal{Y}$. Let a classifier be $f : \mathcal{X} \to \mathcal{C}, \mathcal{C} \subseteq \mathbb{R}^k$.

A loss function $L$ is a map $L : \mathcal{C} \times \mathcal{Y} \to \mathbb{R}^+$. We use $\mathbb{E}$ to denote expectation. Given any loss function $L$ and a classifier $f$, we define the $L$-$risk$ of $f$ by

$$R_L(f) = \mathbb{E}_{\mathcal{D}}[L(f(\mathbf{x}), y_{\mathbf{x}})] = \mathbb{E}_{\mathbf{x}, y_{\mathbf{x}}}[L(f(\mathbf{x}), y_{\mathbf{x}})]. \tag{2}$$

Let $\{(\mathbf{x}, \hat{y}_{\mathbf{x}})\}$ be the mutated data, where

$$\hat{y}_{\mathbf{x}} = \begin{cases} y_{\mathbf{x}}, & \text{with probability } (1 - \eta_{\mathbf{x}}) \\ j \in [k] \backslash y_{\mathbf{x}}, & \text{with probability } \overline{\eta}_{\mathbf{x}_j}. \end{cases}$$

For all $\mathbf{x}$, conditioned on $y_{\mathbf{x}} = i$, we have $\sum_{j \neq i} \overline{\eta}_{\mathbf{x}j} = \eta_{\mathbf{x}}$. The noise is called symmetric or uniform if $\eta_{\mathbf{x}} = \eta$. For uniform or symmetric noise, we also have $\overline{\eta}_{\mathbf{x}_j} = \frac{\eta}{k-1}$. We consider a symmetric loss function $L$ that satisfies the following equation, where $C$ is a constant.

$$\sum_{i=1}^{k} L(f(\mathbf{x}), i) = C, \forall \mathbf{x} \in \mathcal{X}, \forall f. \tag{3}$$

Let $S_\eta = \{(\mathbf{x}_n, \hat{y}_{\mathbf{x}_n}), n = 1, ..., m\}$ be the mutated training data. We call $\eta$ as a mutation degree. Let $f$ be the model trained on $S$, $f_\eta$ be the model trained on $S_\eta$[1]. $f$ and $f_\eta$ are from the same learner (with identical hypothesis space). Let $r(\mathbf{x}) = (L(f_\eta(\mathbf{x}), \hat{y}_{\mathbf{x}}) - L(f(\mathbf{x}), y_{\mathbf{x}}))/\eta$ be the loss change rate between $f$ and $f_\eta$, $\mathbf{x} \in \mathcal{X}$. For uniform noise, we have:

$$\mathbb{E}_{\mathcal{D}}[r(\mathbf{x})] = \mathbb{E}_{\mathcal{D}}\left[\frac{L(f_\eta(\mathbf{x}), \hat{y}_{\mathbf{x}}) - L(f(\mathbf{x}), y_{\mathbf{x}})}{\eta}\right] \tag{4}$$

$$= \mathbb{E}_{\mathcal{D}}\left[\frac{(1-\eta)L(f_\eta(\mathbf{x}), y_{\mathbf{x}}) + \frac{\eta}{k-1}\sum_{i \neq y_{\mathbf{x}}} L(f_\eta(\mathbf{x}), i) - L(f(\mathbf{x}), y_{\mathbf{x}})}{\eta}\right] \tag{5}$$

$$= \mathbb{E}_{\mathcal{D}}\left[\frac{1-\eta}{\eta}L(f_\eta(\mathbf{x}), y_{\mathbf{x}}) - \frac{1}{\eta}L(f(\mathbf{x}), y_{\mathbf{x}}) + \frac{1}{k-1}\sum_{i \neq y_{\mathbf{x}}} L(f_\eta(\mathbf{x}), i)\right] \tag{6}$$

$$= \frac{1-\eta}{\eta}R_L(f_\eta) - \frac{1}{\eta}R_L(f) + \frac{C - R_L(f_\eta)}{k-1} \tag{7}$$

$$= (\frac{1}{\eta} - 1 - \frac{1}{k-1})R_L(f_\eta) - \frac{1}{\eta}R_L(f) + \frac{C}{k-1} \tag{8}$$

$$= (\frac{1}{\eta} - \frac{k}{k-1})R_L(f_\eta) - \frac{1}{\eta}R_L(f) + \frac{C}{k-1}. \tag{9}$$

If we consider $L$ to be error rate[2], $C = 1$. Now consider the situation for multi-class classification problems, we *mutate the labels by label swapping:* to mutate the labels by replacing them with the

---

[1] $f$ is the same as $f(S)$ in Equation 15.

[2] The loss function $L$ does not have to be error rate (0-1 loss). Any loss function that satisfies Equation 3 can be applied. In the present work, we use error rate (also accuracy) considering its popularity.

next label in the label list, the final label in the label list is replaced with the first label in the label list. In this way, we have $\overline{\eta}_{\mathbf{x}_n j} = \eta$. Thus, Equation 9 becomes:

$$\mathbb{E}_{\mathcal{D}}[r(\mathbf{x})] = (\frac{1}{\eta} - 2)R_L(f_\eta) - \frac{1}{\eta}R_L(f) + 1. \tag{10}$$

Thus, we have:

$$R_L(f) = (1 - 2\eta)R_L(f_\eta) - \eta\mathbb{E}_{\mathcal{D}}[r(\mathbf{x})] + \eta. \tag{11}$$

Let $T(f)$ be the accuracy of $f$ over distribution $\mathcal{D}$, $T(f) = 1 - R_L(f)$, $T(f_\eta) = 1 - R_L(f_\eta)$. Let $\widehat{T}(f)$ be the empirical accuracy of $f$ on training data with size $n$: $\widehat{T}(f) = \frac{1}{n}\sum_{i=1}^n 1_{f(x_i)=y_{x_i}}$, where $1_w$ is the indicator function of event $w$. We have:

$$T(f) = (1 - 2\eta)T(f_\eta) + \eta\mathbb{E}_{\mathcal{D}}[r(\mathbf{x})] + \eta. \tag{12}$$

**Empirical Model Validation Measurement**

Equation 12 specifies a theoretical metamorphic relation: Mutation degree $\eta$ defines the relationship between inputs $S$ and $S_\eta$: under each class of $S$, $\eta$ proportion of the labels are mutated with label swapping (see more in Section A.1), yielding $S_\eta$. Such input changes lead to the expected output changes reflected by $T(f)$, $T(f_\eta)$, and loss change rate $r(x)$ in the equation.

The calculation of such a theoretical metamorphic relation, however, is impractical, because data distribution $\mathcal{D}$ is unknown, the expectation calculation is also unrealistic. Inspired by Equation 12, also considering that our motivation is to empirically measure how good a learner fits the available training data, we change the expectations on data distribution $\mathcal{D}$ into empirical observations on the available training data. This leads to the following measurement, $m$, to empirically access a learner[3].

$$m = (1 - 2\eta)\widehat{T}_S(f_\eta) + \eta\hat{r} + \eta \tag{13}$$

$$= (1 - 2\eta)\widehat{T}_S(f_\eta) + \eta\frac{\widehat{T}_S(f) - \widehat{T}_{S_\eta}(f_\eta)}{\eta} + \eta \tag{14}$$

$$= (1 - 2\eta)\widehat{T}_S(f_\eta) + \widehat{T}_S(f) - \widehat{T}_{S_\eta}(f_\eta) + \eta. \tag{15}$$

In Equation 15, $S$ is the original training data, $S_\eta$ is the mutated training data with mutation degree $\eta$ ($\eta \leq 0.5$[4]), $f$ is the model trained on the original training data, $f_\eta$ is the model trained on the mutated data, $\widehat{T}_S(f)$, $\widehat{T}_S(f_\eta)$ are the accuracy of $f$ and $f_\eta$ based on the original training labels, respectively. $\widehat{T}_{S_\eta}(f_\eta)$ is the accuracy of $f_\eta$ based on the mutated training labels.

**Connection between $m$ and Our Intuition:**

Interestingly, Equation 15 matches well with our intuition introduced in Section 2.1. In particular, if the learner is less affected by the mutated labels, the predictive behaviours of the trained model with mutated labels should be closer to that of the model trained with the original labels. This leads to a larger $\widehat{T}_S(f_\eta)$ and $\hat{r}$, as long as the mutation degree $\eta$ is fixed. The matching between $m$ and our intuition provides extra supports for the reliability of MV in validating machine learning models.

The calculation of $m$ can also be regarded as a type of mutation score Jia & Harman (2010); Papadakis et al. (2019) for model validation. As explained in Section 2.1, we expect that a good learner kills more mutants. However, the intuitionistic mutation score (i.e., the proportion of killed mutants) has a limitation in model validation: a poor learner that makes random guesses may also kill many mutants.

---

[3]The purpose of $m$ is not to approximate $T(f)$, but to measure how good a learner fits the available training data. However, if the training data is sufficiently large, $m$ is expected to be close to $T(f)$.

[4]

Equation 15 covers the mutant killing results (i.e., by calculating accuracy decrease rate $r$), but also fixes this limitation by also considering the accuracy on the original correct labels (i.e, $\widehat{T}_S(f_\eta)$). Thus, it can be regarded as a mutation score calculation adapted to suit the model validation scenario.

The larger a learner's $m$ is, the better the learner fits the training data. Thus, we adapt the concept of metamorphic relation in the scenario of model validation and extend it to a qualitative measurement, rather than a simple binary judgement. However, the value of $m$ can also reflect the metamorphic relation we introduced in Section 2.2: let us define that once a learner's $m$ is below a threshold (e.g., 0.8), the metamorphic relation is violated. The violation then indicates a fault in model fitting: the learner is either over-complex (with a large $\widehat{T}_S(f)$) or over-simple (with a small $\widehat{T}_S(f)$) for the training data.

## A.2 INFLUENCE OF MUTATION DEGREE

From Equation 15, it can be seen that the calculation of MV involves a mutation degree, $\eta$, for generating the mutated training data. The value of $\eta$ needs to be fixed during the calculation. However, if Equation 15 is reliable , the influence of $\eta$ on model validation should be minor. This section empirically explores whether this is true.

The first sub-figure in Figure 7 shows the results for UCI datasets. The second sub-figure shows the results for the three large image datasets. It reveals that for most datasets (except for the three smallest datasets, i.e., iris, wine, and heart), the values of MV remain almost identical with different mutation degrees. This is because there is the constant term $\eta$ at the end of Equation 15 when calculating MV, which cancels out the decrease of the detected mutants. This observation provides further evidence for the reliability of our calculation formula shown in Equation 15.

For the three very small datasets, i.e., iris, wine, and heart, with fewer than 300 data points, we observe that a larger noise degree leads to a smaller MV. This may be because label mutations have more influence on very small datasets. Nevertheless, with different mutation degrees, we observe that the effectiveness of MV in model selection and hyperparameter configuration do not change, because the relative rankings of models/hyperparameters remain unchanged. For example, as shown by the third sub-figure in Figure 7, even for the smallest dataset iris, the recommended maximum depths for Decision Tree are identical.

The same as the choice of $n$ in n-fold cross-validation, although we demonstrate that the choice of $\eta$ doest not affect model validation conclusions, there may be a best practice for selecting $\eta$ under different application scenarios. We call for future work and practices to explore this.

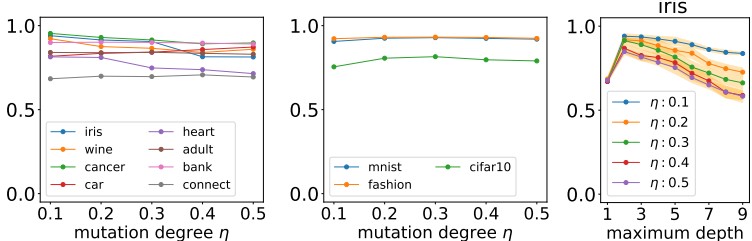

Figure 7: Influence of mutation degree $\eta$ on MV. The results show that different $\eta$ lead to similar MV values and identical model validation conclusions.

## A.3 INFLUENCE OF TRAINING DATA SIZE ON MV

In addition to RQ4, we further investigate what would happen to MV should we deliberately use training data much more than normally expected. That is, we go beyond the assumption that there is no need to increase data when the test accuracy becomes stable. As we can see from Figure 8, MV is more responsive to changes in training data size than test accuracy. For learners that are over-complex to the data, when adding more training data no longer increases test accuracy, MV continues to increase, indicating that the model's resilience to incorrect labels continues to increase. The extra training data improve the robustness of the learner to training label noise. The pattern that MV no

longer changes when model complexity increases is also a signal to developers that the training data is perhaps larger than necessary.

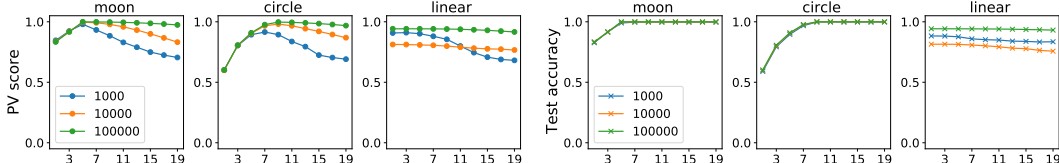

Figure 8: MV (first row) and test accuracy (second row) on training data of increasing size with different maximum depths (horizontal axis). MV is responsive to data size changes; 2) MV no longer decreases for large depths when the training data size is sufficiently large.

## A.4 Efficiency of MV in Model Validation

In this part, we compare the efficiency of MV, CV (3-fold), and validation accuracy (for image datasets), using the synthetic datasets on 7 classifiers (with the same setup in RQ1), the 8 UCI datasets on Decision Tree (with a fixed maximum depth of 5), and the 3 image datasets (with a fixed dropout rate of 0.5, and learning rate of 0.0001). The deep learning experiments with the three image datasets were run on Tesla V100, with 16GB Memory and 61GB RAM.

Table 5 shows the results. For brevity, we show only the results for the moon synthetic datasets. Overall, both MV, CV, and validation accuracy have good efficiency on these datasets. Note that in practice developers often use 5-fold or 10-fold CV, which have larger time cost than the 3-fold CV.

The cost of MV mainly comes from data mutation and model training. As observed from Table 5, larger datasets take more time to get MV values. Our results demonstrate that the cost of MV is manageable and comparable to 3-fold CV and validation accuracy in both classic learning and deep learning. In particular, for the three large image datasets, MV costs only half the time of 3-fold CV.

What is more, as demonstrated by RQ1 and RQ2, the effectiveness and stability of MV help to conduct model selection and hyperparameter configuration more quickly. Thus, it helps to save cost in selecting the best learners for a given training set. On the other hand, MV is sensitive to over-complex models, and can help to select the simplest model with reasonable test accuracy. This will also reduce the model training and maintainability cost in the long run.

Overall, MV has comparable efficiency to 3-fold CV and validation accuracy. For the three deep learning tasks, MV's efficiency doubles that of 3-fold CV.

| Dataset | Learner | MV-time | CV-time | Dataset | Learner | MV-time | CV-time |
|---|---|---|---|---|---|---|---|
| moon | Linear SVM | 0.002s | 0.003s | iris | Decision Tree | 0.021s | 0.003s |
| moon | RBF SVM | 0.003s | 0.004s | wine | Decision Tree | 0.004s | 0.007s |
| moon | Gaussian Process | 0.161s | 0.157s | cancer | Decision Tree | 0.023s | 0.017s |
| moon | Decision Tree | 0.001s | 0.002s | car | Decision Tree | 0.011s | 0.014s |
| moon | Random Forest | 0.046s | 0.064s | heart | Decision Tree | 0.004s | 0.008s |
| moon | AdaBoost | 0.124s | 0.185s | adult | Decision Tree | 0.386s | 0.385s |
| moon | Naive Bayes | 0.002s | 0.003s | bank | Decision Tree | 0.118s | 0.109s |
| | | | | connect | Decision Tree | 0.195s | 0.196s |
| **mean** | – | **0.048s** | **0.060s** | – | – | **0.095s** | **0.092s** |

| Dataset | Learner | epoch | MV-time | CV-time | Validation-time |
|---|---|---|---|---|---|
| mnist | convolutional neural network | 10 | 2.772min | 5.296min | 1.761min |
| fashion-mnist | convolutional neural network | 10 | 2.803min | 5.352min | 1.778min |
| cifar10 | convolutional neural network | 50 | 12.052min | 24.315min | 8.072min |
| **mean** | – | – | **5.876min** | **11.654min** | **3.870min** |

Table 5: Efficiency of MV, CV, and validation accuracy. The top/bottom sub-table shows the results for classic/deep learning. We observe that the efficiency of MV is comparable to that of 3-fold CV and validation accuracy.

## A.5 MODEL SELECTION WITH UCI DATASETS (RQ1)

In this part, as an extension to RQ1, we present the results of model selection using UCI datasets. Note that we do not use test accuracy as the ground truth for model selection considering the possible limitations of test accuracy we discussed in the introduction. Instead, we present the results to demonstrate how MV differs from CV and test accuracy, as well as how it complements CV and test accuracy in model selection when developers observe similar accuracy results across different learners.

Figure 9 shows the results. It is interesting that we do have observations on MV that are consistent with common machine learning knowledge. In particular, for the four smallest dataset (i.e., iris, wine, cancer, and heart), MV suggest the two simplest learners (i.e., Linear SVM and Naive Bayes).

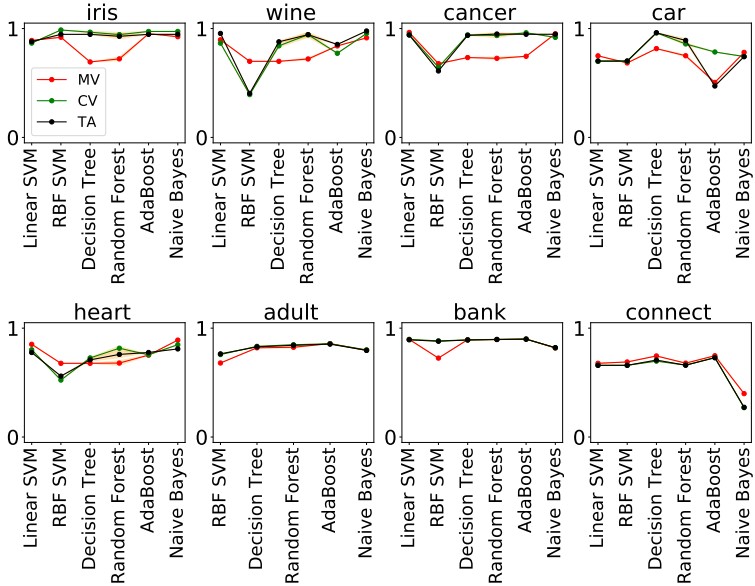

Figure 9: Model selection results with UCI datasets (extended analysis for RQ1).

### A.6    MODEL SELECTION WITH RANDOM LABEL REPLACEMENT

In the main body of this work, we present empirical results with MV calculated using label swapping. In this part, we further explore the effectiveness of another label mutation approach: random label replacement. That is, when conducting label mutation, we replace the original label with a label that is randomly chosen from the label list. We compare the performance of these two label mutation approaches in model selection with the ground truth provided by synthetic datasets. Figure 12 shows the results. We observe that random label replacement is less accurate than label swapping in model selection, but is still more accurate than CV and test accuracy.

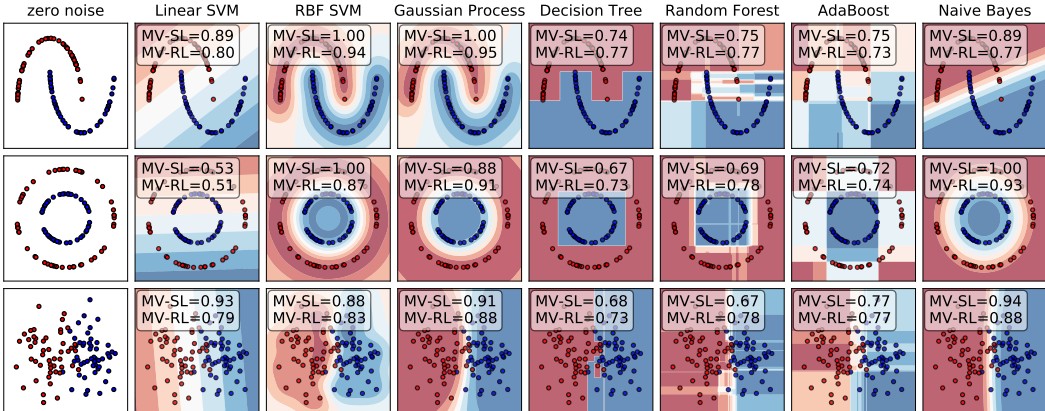

Figure 10: Performance of MV in model selection with label swapping (MV-SL) and random label replacement (ML-RL).

### A.7

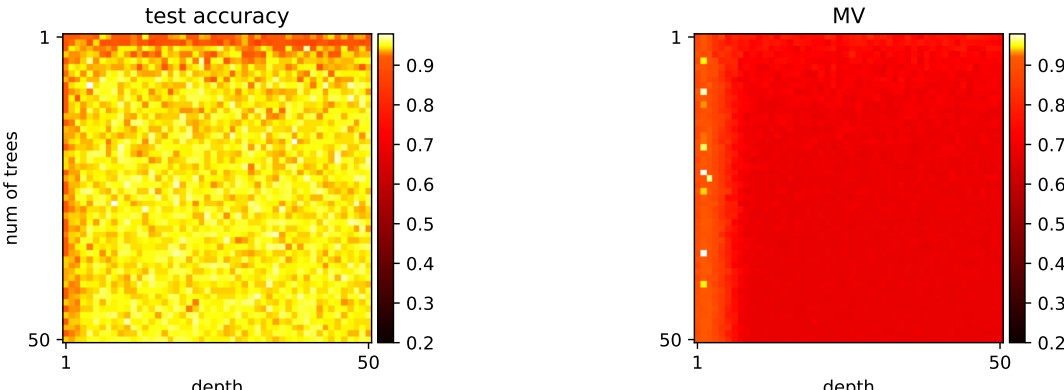

Figure 11: Performance of MV in suggesting hyperparameters that follow Occam's Razor on dataset Cancer. The training data has only 300 samples. The low values of MV on large depths and number of trees provide warnings to developers that the hyperparameters are over complex and violate the rule of Occam's Razor. The unnecessary complexity in the complex learner affects the interpretability of the learner, also making it vulnerable to training label attacks.

A.8

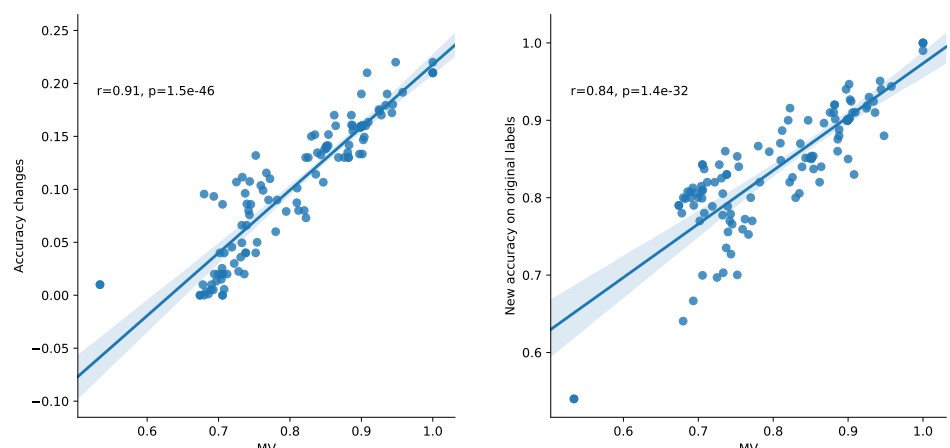

Figure 12: Correlation between MV, training accuracy changes, and the new training accuracy based on the original labels.

