# OpenReview forum: "Model Validation Using Mutated Training Labels: An Exploratory Study"
_ICLR.cc/2022/Conference — ICLR 2022 Submitted_

### Official Review · Reviewer_1Tm6 · 2021-10-25

**Correctness:** 4
**Technical Novelty And Significance:** 3
**Empirical Novelty And Significance:** 4
**Recommendation:** 6
**Confidence:** 3

**Main Review:**

Strengths

The paper proposes an interesting new method for model validation/selection, which may be especially useful in low data regimes in which CV partitions a fraction of the training data to use as a validation set, whereas MV can use the entire training set.

As far as I know, this work is original, and applies techniques of Mutation Testing and Metamorphic Testing from software engineering to develop MV.

The proposed measurement itself is simple, easy to understand, seems to be technically sound, and has a justifiable theoretical foundation.

The paper is well-written and easy to read and follow.

Experiments are extensive on 12 real-world datasets, with comparisons to relevant baselines. The paper also characterizes how MV changes in response to changes in data size, and the fraction of examples mutated.

Weaknesses

I think my main concern is how to choose the fraction of training examples to mutate. It seems mutating large fractions of the data (e.g., 0.4, 0.5) would lead to a significant decrease in the detection of mutants. Thus, is there a suggested optimal value for this parameter, or characterization that can help ML practitioners better decide what value to use?

Why compare against 3-fold cross-validation instead of 5-fold (the default in scikit-learn: https://scikit-learn.org/stable/modules/generated/sklearn.model_selection.GridSearchCV.html?highlight=gridsearch#sklearn.model_selection.GridSearchCV)?

Why did the authors choose the mutation protocol to be choosing the next label in line, instead of selecting a different label uniformly at random; would the results be significantly different if using the latter?

MV only works for classification tasks; can MV be adapted for regression tasks?


Minor Weaknesses

Section 2.2, par. 3: consider using A(S) to represent a model trained on the data subset S; or, rephrase to say f(S) is the OUTPUT of the model for the subset S.

Is AUC used as the performance metric for the binary-classification tasks? AUC is a more sensitive measure of model performance and may lead to different model selections especially for methods such as CV.


Additional Questions

How does mutating training labels compare with mutating feature values? Can the mutation of both lead to a better measurement of model validation/selection?

How does MV perform in the context of model generalization, especially when tuning multiple hyperparameters?






**Summary Of The Paper:**

This paper proposes a new form of model validation called Mutation Validation (MV), which analyzes how well a model fits the training data by first training two models: one using the original training labels, and another using training labels in which a fraction of them (<= 0.5) have been mutated (i.e., the next available class label is used); then, a goodness-of-fit measure is computed via the accuracy of these models on the mutated and non-mutated subsets of the training data, with the idea that the model trained using mutated labels should have similar predictive performance to the original model if less affected by the mutated labels. The paper provides a thorough empirical comparison to other model-validation strategies such as cross-validation (CV) and using a validation/test set; the results suggest that MV provides a more sensitive response to changing hyperparameter values and can better detect the unnecessary complexity of the model than any of the other methods. They also find the value of MV tends to increase with addition of more data, signifying a resilience to mutated labels with additional examples; and they show that MV is relatively robust when choosing what fraction of training examples to mutate.

**Summary Of The Review:**

The paper proposes a new method for model validation/selection which is simple, makes use of ALL the training data, and appears to be more sensitive to overly complex models; this ultimately leads to simpler models via a more robust selection of hyperparameter values that better fit the true underlying data distribution. My main concerns are how to choose an appropriate fraction of the training examples to mutate, and how this method can be adapted to regression tasks.

---

> ### Author Response · Authors · 2021-11-17
> **Response to Reviewer 1Tm6**
>
> > I think my main concern is how to choose the fraction of training examples to mutate. It seems mutating large fractions of the data (e.g., 0.4, 0.5) would lead to a significant decrease in the detection of mutants. Thus, is there a suggested optimal value for this parameter, or characterization that can help ML practitioners better decide what value to use?
>
> **Response**: Our initial experiments shown in Appendix A.2 demonstrate that different mutation degrees lead to very similar MV measurement results. This is because there is the constant term eta at the end of Formula (1) when calculating MV, which cancels out the decrease of the detected mutants.
>
> **Revision**: Thanks for the comment. We have added more explanation in Appendix A.2.
>
> > Why compare against 3-fold cross-validation instead of 5-fold?
>
> **Response**: In our experiments, 3-fold cross validation and 5-fold cross validation have almost identical results. We choose to use 3-fold cross validation because it has lower cost, thereby being more conservative to compare with when studying the efficiency of MV.​​
>
> **Revision**: We added the explanation in Section 3.
>
> > Why did the authors choose the mutation protocol to be choosing the next label in line, instead of selecting a different label uniformly at random; would the results be significantly different if using the latter?
>
> **Response**: We choose the next label in line because it is supported by theory, as shown in Appendix A.1.
>
> **Revision**: We added the results with random label replacement in Appendix A.6. We observe that selecting labels uniformly at random is less accurate than selecting the next label in line in model selection, but is still more accurate than CV and test accuracy.
>
> > MV only works for classification tasks; can MV be adapted for regression tasks?
>
> **Response**: Thanks for this interesting question! MV has the potential to work for regression tasks, where we could mutate data points into obvious outliers and observe how the regression model reacts to it. A proper model is expected to be less affected by the outliers in the training data. We will explore more in the future.
>
>
> >Section 2.2, par. 3: consider using A(S) to represent a model trained on the data subset S; or, rephrase to say f(S) is the OUTPUT of the model for the subset S.
>
> **Revision**: Thanks! We have revised the paper as suggested.
>
> > Is AUC used as the performance metric for the binary-classification tasks? AUC is a more sensitive measure of model performance and may lead to different model selections especially for methods such as CV.
>
> **Response**: AUC is not currently used as a metric for MV. We focus on accuracy for the moment because we notice that it is the most widely reported metric in machine learning publications; there is also theory support as shown in Appendix A.1. We plan to explore more in the future.
>
> > How does mutating training labels compare with mutating feature values? Can the mutation of both lead to a better measurement of model validation/selection?
>
> **Response**: Mutating labels certainly lead to incorrect labels. We can judge a learner based on how it reacts to these incorrect labels in the training data. However, this is not true for mutating features,  where a change in a feature may just yield a piece of valid unseen data. Thus, we choose to mutate labels in MV.
>
> > How does MV perform in the context of model generalization, especially when tuning multiple hyperparameters?
>
> **Response**: There is no ground truth for model generalization with real data. Developers often use hold-out test sets to simulate future unseen data. In this context, our experiments in Figure 5 show that most of the times MV has identical hyperparameters tuning results as validation and test accuracy. When MV and test accuracy have different key influence points (i.e., on cifar10 (0.4 v.s. 0.2)), it is because there is a big capacity jump between 0.2 and 0.4. The result suggests that the optimal dropout rate with comparable validation/test accuracy is between 0.2 and 0.4. Thus, MV complements CV and test accuracy to help developers to find the best hyperparameters with comparable test accuracy.

---

> > ### Comment · Reviewer_1Tm6 · 2021-11-22
> > **Response**
> >
> > I have read and thank the authors for their response and intend to increase my score to 8.

---

### Official Review · Reviewer_S95g · 2021-11-02

**Correctness:** 2
**Technical Novelty And Significance:** 2
**Empirical Novelty And Significance:** 3
**Recommendation:** 3
**Confidence:** 4

**Main Review:**



The paper is well written and well motivated, and proposes a simple
method for model validation, mutation validation (MV), based on
changing the labels of training instances, and measuring how well the
learner fits to the new mutated problem, and combining the original and new
accuracies. Variety of experiments are conducted to show the potential
of the approach to complement traditional techniques such as
cross-validation.

However, the major shortcoming of the paper is performing the model
selection experiments on only toy problems with 2 features.  The
method is simple enough (and elegant!)  that I implemented the MV score and compared a
few learners on a couple of the authors' UCI datasets (breast cancer
and Bank) and other datasets.  I double checked my implementation and
matched the scores and trends with what the authors have published
(more details below). I found that unfortunately, the MV score can
pick simple depth-limited decision tree models vs bag of trees (of
larger or unlimited depth, so  learners with higher fit capacity) while bag
of trees has a significant higher test accuracy (simple 10-fold
validation tends to pick the bag of trees).

It seems to me model selection is the main potential application of
MV, specially because unlike cross-validation, MV doesn't give you a
sense of performance on test (beyond training performance). However, if it can fail in these simple
scenarios, the impact of this work would be very limited (the other attributes
of MV, such as stability, shown in subsequent experiments in the paper would be minor if it doesn't do a good job
at model selection...)

More investigation is required, and I am not sure if this shortcoming
can be remedied.  This observation suggests MV has an undue bias
towards simpler models, or is not appropriate for comparing across
learning algorithms, or fails with committees/ensembles, etc.


Suggestion for model selection experiments:

The model selection experiments are done on toy problems in the paper. I understand
the 'true model' is unknown in the real datasets (and one can get  certain
insights into algorithm behavior with synthetic data), but what we
care about is test/generalization accuracy any way in practice (in a
simplest/standard application of supervised machine learning). The
authors can keep that section on toy problems (shortened) but also
include a variety of publicly available datasets (UCI, etc) like they
did for the other subsections. To evaluate model selection, keep some
portion for test in each trial (20%, 50%, 80%), and compare
head-to-head say 10-fold validation with 10 trials of MV to select a
model (a couple or more learning algorithms) on the training
portion. How often does MV beat 10fold validation, in selecting the
better model, ie the one with lower test set error, on a particular
problem, as well as across problems?  (of course, both the learning
algorithms to choose from and problems should have some diversity)

--------

More details of my experiments:

The proposed MV measure is simple and elegant, and it was simple enough
that I implemented it and compared it to say 10-fold cross-validation,
head to head on the breast cancer and Bank (binary class datasets, as I didn't implement
the authors' label shifting method for multiclass ). I
saw a similar result on another dataset (text classification).  I'll
focus on breast cancer.

I first matched scores from my implementation (MV and
cross-validation) and validations with the author's results on the
breast cancer dataset (Figure 3, eg 3rd plot is for Wisconsin breast-cancer).
So a single decision tree gets a MV score of around
0.93 (eta of 0.2 as in paper) and that's the peak of MV for breast-cancer. Higher depths
progressively lead to lower MV scores.

Unfortunately MV fails in model selection, when comparing decision
trees of say depth 3 to say bagged trees (say 50 trees of depth 20,
each trained on a bootstrap sample of training): The bag has a better
 accuracy on the test split (generalization), but the MV method substantially
favors a single tree of small depth.  The MV scores of the bag of
trees (in 0.6 to 0.8 ) are not even close to single trees of small
depth (around 0.9 and higher).  For instance, when we random split the
~600 instances of this data into two halves (half for training, half for
test) in each of 50 trials, 10-fold cross-validation picks the model
with smaller error ~40 times, while MV picks the better model only ~3
times (both working on the training split to pick between the two models,
a tree of depth 3 vs a bag of trees). The average over trials of
01-error of a single tree is 0.076, while for the bag it's
0.055 (significantly better). Changing the training portion or the eta of MV, etc,
does not change this comparison much.

There are 3 elements in the MV score: the training accuracy (on
original unmuted data), and the accuracy of the model trained on muted
data (some labels flipped) on the same (muted) sample it was trained
on and on original data.  The issue is that the bagged tree gets a
high training accuracy on a mutated sample (such as near 1.0), while the
accuracy on original labels either does not improve or goes down, so
the MV score of the committee of trees, which punishes for fitting to
muted data, goes down substantially, while a single tree of small
depth has a different accuracy profile, in particular lower accuracy
on the mutated sample, so its overall MV score remains relatively
high.  And it may be that a simple combination of these accuracy
scores is not sufficient for successfully picking a model in practice.


The above observation suggests MV as currently developed has an undue bias towards simpler models,
or is not appropriate for comparing across learning algorithms, or
fails with committees/ensembles, etc. To me, currently, the MV score
has most potential (possible advantage over cross-validation) for
parameter selection for a fixed learning algorithm, but that also
requires more careful and extensive investigation.

Another (secondary) issue is class imbalance (eg the Bank dataset is somewhat
imbalanced at ~10% positive). MV, as developed in this work, is
appropriate when 01-error is useful. As class imbalance grows (for
binary-class problems), the utility of MV further degrades.



-------------------------

(My review before reviewer discussion and experiments)

I enjoyed reading the paper!

The paper is  written clearly and provides good motivation,
examples/illustrations, and the techniques are understandable and
novel to best of my knowledge, and I expect very practical too.
Especially, with deep neural networks with high capacity to fit, these
new validation techniques are timely.

---------------

Detailed feedback/recommendations (some comments or suggestion
are based on my reading the paper linearly!  ):

pg 1. minor: What is "mutating a program' mean, in a bit more detail?
in 'Mutation testing mutates the program, then re-executes the tests
to monitor ' ...  a few additional words of description (eg to get sense of
the extent of change), such as 'it's one or a few
lines of code change' (or short examples of code mutating could help
here)..  (however, changing labels and changing code are pretty
understandable)..

pg 1. metamorphic relations is more ambiguous (than mutation) at this
point: "metamorphic relations, which is the relationship between input
changes and output change" but later  on pg3 you do describe it more
clearly ('Metamorphic relation specifies how a change in the input
should result in a change in the output'). Could repeat that point
here too in giving a quick description of metamorphic for better
clarity. (possibly replacing the more general ambiguous description
"which is the relationship between..")


pg1.  Perhaps replace 'Furthermore, ' by a contrasting phrase such as
'On the other hand', in 'Furthermore, an over-simple learner ..'


section 1, pg2. Perhaps mentioning the (potential) short comings of MV
would be useful when you list the advantages of MV at this point: what
are potential shortcomings as compared to the typical validation
techniques?  (at this point in the paper, without reading the rest of
paper, I can see it requires a parameter for binary classification,
ie what fraction of labels to flip, but other techniques also require
deciding on portion to set aside.. and efficiency of the test could be
an issue specially if the flip fraction needs to be varied
.. ).. Perhaps the most notable is that it doesn't give you the
out-of sample accuracy, \ie a sense of generalization
performance/accuracy. How about other theoretical/conceptual potential
drawbacks?

pg3:  perhaps replace 'measurement' in 'MV measurement' with 'score'
(so 'MV score' ) to imply to reader that a higher value is better.

pg3. Can eta be above 0.5 for the multiclass case.  perhaps a very brief discussion
would be useful.

Experiments:

-- why eta of 0.2 (mutation fraction)?  How did settings do, such as
 0.1 and sensitivity to it?  Of course, it's good to see that eta of
 0.2 was uniformly adequate.  I think you can point to the appendix at
 this point (later I found the appendix to have a section! and earlier
 in the paper, you do state that the effect of eta is studied in
 appendix. Probably it's better to mention it here.).

-- Maybe a better name for label swapping is "label shifting"?

-- Why this specific manner of label flipping for multiclass cases (it's intriguing!),  "label
 swapping"? ie to replace a label with the next label in the label list"?
 Why not just randomly pick a label from other classes? (or pick based
 on class proportion, etc..)  this was the one puzzling aspect of the
 experiments!  since the first (simplest) idea that comes to mind is
 random selection from available classes (of course, this question is
 relevant only to the non-binary problems you looked at). It appears
 this label swap generates systematic error, not random noise, but I
 am not sure what the implications are (eg could be easier, than
 random label noise, to fit the training data, since there is a
 pattern to the injected label errors, but generalization degradation
 can be impacted further by this, vs random noise.. ) Later I saw in
 the appendix that the theoretical result from the appendix motivates
 this choice vs simpler random noise. it would be good to mention it
 when you state you are using label swapping, vs uniform random
 .. Also, it would be good to empirically see whether label swapping
 (or "label-shifting") vs random uniform selection of labels, makes a
 difference. (I don't think you currently do this comparison, checking the appendix)

* The scores in figure 2 specially are hard to see/read.

pg 9.

insert 'more' in "We show that MV is effective and stable than the
currently adopted CV, validation accuracy, and test accuracy"

pg 9: (in contrasting with noise injection) perhaps better to replace
'inputs' with 'features', eg. in "2) mutates labels, not inputs" (labels
are input too, but agreed, labels are inputs to the learner not the model)

pg 9: I am not sure if 'model robustness' does not include  or highly overlap with model
fitting.. (it's a general phase) in 'aims to validate model fitting,
rather than model robustness.' perhaps make model robustness more
specific by adding something like 'robustness to feature
perturbations'


Appendix. Minor: m is used as training set size at one point (while in the rest of the paper, it's the mv score).

**Summary Of The Paper:**


The authors propose and explore  a method based on changing (mutating) the labels of training data to do model selection (
selecting among learning
algorithms, or hyper parameters, etc).  The method is very practical and a number of experiments show the properties of the approach
and its potential to complement  cross validation and other validation techniques.

**Summary Of The Review:**


The paper is clearly written and well structured, with good
motivation, examples/illustrations, and the techniques are
understandable and novel to the best of my knowledge.  The work
is mainly empirical, but the experiments need to be substantially
extended, and the basic approach may need further development,
to show the benefits of mutation validation.

---

> ### Author Response · Authors · 2021-11-17
> **Response to Reviewer S95g (1/2)**
>
> > The major shortcoming of the paper is performing the model selection experiments on only toy problems with 2 features (in RQ1).
>
> **Response**: Thanks for your comments. In RQ1, we choose to use synthetic datasets and avoid using real datasets for two reasons. First, these synthetic datasets are used by SKlearn to demonstrate model selection, and are well known to developers. Second, since test accuracy can be unreliable, these synthetic dataset provide us known ground-truth decision boundaries for validating model selection.
>
> **Revision**: Having said that, we agree that including real datasets for model selection could provide extra information to evaluate MV. We have updated our paper with real datasets results as well as the analysis for RQ1 in the appendix, as required.
>
> > To evaluate model selection, keep some portion for test in each trial (20%, 50%, 80%), and compare head-to-head say 10-fold validation with 10 trials of MV to select a model (a couple or more learning algorithms) on the training portion. How often does MV beat 10-fold validation, in selecting the better model.
>
> **Response**: Thanks for your suggestion. Please note that the motivation for MV is the limitations in out-of-sample validation (including test accuracy) as we listed at the beginning of the introduction. Indeed, as shown by Figure 2 and Table 2, test accuracy does not work well for model selection even with very simple data. We also frequently hear complaints from developers in larger IT companies about the unreliability of offline test accuracy. For these reasons, we did not choose test accuracy as the ground truth for model selection.
>
> However, as we said in Section 5.2, despite these limitations, test accuracy is the most important model validation method in practice before online model deployment. This paper does *not* aim to propose MV to replace cross validation or test accuracy. Instead, we provide the first exploratory study on the possibility of using MV *as a complement* of cross validation and test accuracy.
>
> > Unfortunately MV fails in model selection, when comparing decision trees of say depth 3 to say bagged trees (say 50 trees of depth 20, each trained on a bootstrap sample of training): The bag has a better accuracy on the test split (generalization), but the MV method substantially favors a single tree of small depth. The MV scores of the bag of trees (in 0.6 to 0.8 ) are not even close to single trees of small depth (around 0.9 and higher)...The above observation suggests MV as currently developed has an undue bias towards simpler models.
>
> **Response**: Yes, you are right that MV and test accuracy do not always agree. We discussed such disagreements in Section 4.2 and 5.2. In the case you mentioned, for the 50-tree-20-depth learner, its MV is around 0.68, its test accuracy is around 0.95; for the single-tree-3-depth learner, its MV is around 0.89, its test accuracy is around 0.93. This is a warning that the 50-tree-20-depth learner violates the rule of Occam’s Razor; the complexity of the most optimal learner for the training data (with only 300 samples) is between that of these two learners. Our experimental results *shown in Appendix A.7* in our revision confirm that for the 300-sample training data, there is definitely no need to use 50 trees with depth of 20 to get a test accuracy of 0.95. The unnecessary complexity in the 50-tree-20-depth learner affects the interpretability of the learner, also making it vulnerable to training label attacks.
>
> Our experimental results contain many examples against the claim “MV has an undue bias towards simpler models”. For example, in Figure 2, for the moon and linear data distribution with zero noise, MV favors Random Forest over Decision Tree. In RQ2, as the learners become more and more complex for various datasets, MV is demonstrated to first increase, then decrease.
>
> **Revision**: We added Appendix A.7 to show how MV complements test accuracy to find simper models with appropriate test accuracy.

---

> > ### Comment · Reviewer_S95g · 2021-11-30
> > **I thank  the authors for their  paper improvements and responses**
> >
> > Summary: I thank the authors for their response and enhancements to the paper.   However,  I  intend to keep my original  recommendation, as explained below.
> >
> > Main issue: I still think the utility of MV is much more nuanced and requires more investigation. In particular, the claim in the abstract that
> > "Our results demonstrate that MV is accurate in model selection: the model recommendation hit rate is 92% for MV and less than 60% for out-of-sample- validation."  should be modified.
> >
> > * Note that an improvement of 1% or 2% in accuracy on test set as long as significant should not be attributed to overfitting.. (if you look
> > at as relative reduction in error, it may be not just significant but also substantial )
> >
> > * At this point, it seems that if one expects the problem to have class noise, then perhaps model selection via the help of MV (which simulates additional label noise) could be useful.
> >
> > * A case where MV fails (compared to plain cross-validation):   Imagine a binary learning problem where a good portion of the predictive (boolean) features  have low support, ie, these features  appear (have value 1) say once or a few times but always in the positive or always in the negative instances (and such instances are not covered by other predictive features).   There may also exist a few predictive features with high support,
> > but picking only these high support features  fall far short of covering all the instances of either class.
> > Then a classifier (a learner)
> > that picks all the predictive (low and high support) features ( a deep decision tree ) does well (low test error, achieves high recall), compared to one that only picks  high support (evidence) features (achieving low recall ), but MV punishes the classifier  that picks the low-support features or at best is indifferent to it (under the added label noise such a learner may pick bad features ).     This scenario happens for example in text classification (with a long tail of features or words).    If there is little or no label noise, MV can unnecessarily punish classifiers that pick a lot of (low support) features (but the learning task may require that).    In this sense, MV can be unnecessarily  conservative.
> >
> > * My comment on class-imbalance:  assume the positive class is only < 1% of the instances. Then accuracy does not show sufficient sensitivity for comparing models. The classifier that always says NO gets 99% accuracy (but has 0 recall).  MV, being based on accuracy, may also fail in picking the classifier with the best combination of precision and recall.
> >
> > Recommendations:
> >
> > * Please quantify the performance of cross-validation vs.  MV  on the real datasets (with statistical significance). Not just pictures.
> >
> > * Explicitly experiment with label noise on real datasets.   MV  may show its utility in such scenarios.

---

> > > ### Author Response · Authors · 2021-11-30
> > > **Clarification on misunderstandings**
> > >
> > > Thanks for the reply.
> > >
> > > >Note that an improvement of 1% or 2% in accuracy on test set as long as significant should not be attributed to overfitting.. (if you look at as relative reduction in error, it may be not just significant but also substantial.
> > >
> > > We agree that a minor improvement of test accuracy is important. MV does not punish complexity but punishes over-complexity. If model A has 2% larger test accuracy but lower MV score than model B, it is a warning for developers to find model C which is simpler but has similar test accuracy as model A.
> > >
> > > We'd also like to repeat that MV **does not** aim to replace cross-validation nor test accuracy. Instead, we provide the first exploratory study on the possibility of using MV as a **complement** of cross-validation and test accuracy.
> > >
> > > >At this point, it seems that if one expects the problem to have class noise, then perhaps model selection via the help of MV (which simulates additional label noise) could be useful.
> > >
> > > Please note that our results shown by Figure 2 **with** and **without** class noise demonstrate that MV is useful in model selection in both scenarios.
> > >
> > > >MV punishes the classifier that picks the low-support features or at best is indifferent to it (under the added label noise such a learner may pick bad features ).
> > >
> > > On the contrary, if under the added label noise the learner picks bad features, it will yield low performance, leading to a large performance decrease and a large MV score. Our theory support in the Appendix shows that MV is not affected by feature distribution. We just did some experiments on data samples. The results also demonstrate that there is no need to worry about the case you mentioned.
> > >
> > > Thanks for the comment. We will discuss this in our next version.

---

> ### Author Response · Authors · 2021-11-17
> **Response to Reviewer S95g (2/2)**
>
>
> > Another (secondary) issue is class imbalance (eg the Bank dataset is somewhat imbalanced at ~10% positive). MV, as developed in this work, is appropriate when 01-error is useful. As class imbalance grows (for binary-class problems), the utility of MV further degrades.
>
> **Response**: MV is *NOT* affected by class imbalance, which can be supported by its theory inspiration shown in Appendix A.1.
>
> **Revision**: Thanks for your comment. We have made this clearer in Section 3 in our updated version.
>
> > minor: What is "mutating a program' mean, in a bit more detail?
>
> **Response**: Mutating a program means injecting faults in the program, such as change ‘>’ into ‘<’, change ‘i++’ into ‘i--’, or remove a statement.
>
> **Revision**: We have explained more, as suggested.
>
> > Can eta be above 0.5 for the multiclass case. perhaps a very brief discussion would be useful.
>
> **Response**: Thanks. As introduced at the end of Section 2, eta ranges between 0 and 0.5.
>
> **Revision**: We have added a discussion in Appendix A.1 in the revision, as suggested.
>
> > The scores in figure 2 specially are hard to see/read.
>
> **Revision**: Following your suggestion, we have enlarged the fonts in Figure 2 in the revision to make the figure more readable.
>
> > It would be good to empirically see whether label swapping (or "label-shifting") vs random uniform selection of labels, makes a difference. (I don't think you currently do this comparison, checking the appendix)
>
> **Revision**: Thanks for the comment. Following your suggestion, we have added the results with random label replacement in Appendix A.6. We observe that random label replacement is less accurate than label swapping in model selection, but is still more accurate than CV and test accuracy.
>
> > Minor comments:
>
> **Revision**: Thanks for all these helpful suggestions! We have revised the paper accordingly addressing each point you mentioned.

---

### Official Review · Reviewer_FmyF · 2021-11-02

**Correctness:** 3
**Technical Novelty And Significance:** 4
**Empirical Novelty And Significance:** 4
**Recommendation:** 8
**Confidence:** 4

**Main Review:**

Strengths:
-	A novel approach in the area of validation techniques, which are very conservative
-	Huge empirical validation supported by theoretical basis (see Appendix)
-	Large potential for future studies

Weaknesses:
-	The method looks highly dependent on a particular evaluation metric (performance of a model): for instance, the paper address one case about accuracy (the formula (1), arguments in Sec.2.2, theory in Appendix). It is not clear how to develop a measurement m (analogue of (1)) for other performance techniques
-	Overfitting is a very important problem in the industry. The novel approach requires exploration in “real life” (huge industry datasets and models).
-	Minors:
1)	Fig.1 “A better learner is less”: Which ones are better learners?
2)	In Sec.1, 4th paragraph: “The model recommendation hit rate for MV is 92%” model recommendation hit rate is not a clear term for introduction. Can be clarified?
3)	In Sec.1, 4th paragraph: “set, the average variance is 0.004 and 0.008” Is it reasonable to talk about var among 5(?) runs?
4)	Sec.3, RQ1: how do we define the effectiveness?


**Summary Of The Paper:**

This paper introduces new approach for validation of ML models which is based on top of ideas imported from software development & QA (so-called mutation and metamorphic techniques). The core claimed advantage consist in the ability to avoid cross-validation and validation and test sets (the famous approach to avoid overfitting while learning a ML model). The work contributes also a huge empirical study with experiments for dozens of datasets and several models and hyperparameter tuning.

**Summary Of The Review:**

Nice and clever work. I vote for acceptance due to the “fresh blood” in the so conservative area of validation techniques. This work represents a first empirical study in this new direction, and a lot of future work/studies seem to be followed. So, a lot of things would be nice to check and study to get comprehensive overview of the novel approach, but this first step is enough to be presented at ICLR 2022.

---

> ### Author Response · Authors · 2021-11-17
> **Response to Reviewer FmyF**
>
> > The method looks highly dependent on a particular evaluation metric (performance of a model): for instance, the paper address one case about accuracy (the formula (1), arguments in Sec.2.2, theory in Appendix). It is not clear how to develop a measurement m (analogue of (1)) for other performance techniques.
>
> **Response**: Thanks for the comment. We acknowledge that this is a limitation of MV in the current work. We focus on accuracy because we notice that it is the most widely reported metric in machine learning publications. We plan to explore more in the future.
>
> > Overfitting is a very important problem in the industry. The novel approach requires exploration in “real life” (huge industry datasets and models).
>
> **Response**: Indeed, we often hear complaints about overfitting and the failure of test accuracy in exposing the overfitting from developers in a large IT company. MV is designed to be a practical tool. We look forward to seeking collaboration with machine learning engineers to apply MV in large datasets once the paper is accepted. Thanks for the comment!
>
> > Fig.1 “A better learner is less”: Which ones are better learners?
>
> **Response**: The learners that are neither over complex nor too simple for the training data.
>
> > In Sec.1, 4th paragraph: “The model recommendation hit rate for MV is 92%” model recommendation hit rate is not a clear term for introduction. Can be clarified?
>
> **Response**: Thanks for the suggestion. The hit rate is the ratio of recommended models that match the ground truth model.
>
> **Revision**: We have clarified in Section 4.1 as suggested.
>
> > In Sec.1, 4th paragraph: “set, the average variance is 0.004 and 0.008” Is it reasonable to talk about var among 5(?) runs?
>
> **Response**: Thanks! We have added results for more runs.
>
> > Sec.3, RQ1: how do we define the effectiveness?
>
> **Response**: Sorry for the ambiguity. The effectiveness is how good MV is in recommending learners that best match data patterns.

---

> > ### Comment · Reviewer_FmyF · 2021-11-26
> > **I have read and thank the authors for their response.**
> >
> > Thank you for the response.

---

### Official Review · Reviewer_11Jy · 2021-11-03

**Correctness:** 2
**Technical Novelty And Significance:** 3
**Empirical Novelty And Significance:** 2
**Recommendation:** 5
**Confidence:** 4

**Main Review:**

There are several things to like about the paper -

Novelty-

(1) I think the idea presented in the paper is novel and interesting.

(2) I liked the four research questions posed by the paper in section 3 to validate the usefulness of MV.

Clarity-

(1) I found that the paper was easy to read

(2) The synthetic experiments in Figure 2 provide an intuitive explanation for the algorithm at known distributions.

I think there several shortcomings detailed as follows -

Quality and Significance -

(1) Evaluation: I did not find the experimental section to be convincing enough. For example - consider Figure 3 in the paper where MV has an increasing and a decreasing trend as one increases the depth of decision trees. The paper mentions here that the decreasing trend is due to overfitting and the existing validation methods like CV show no change.  However the claim of overfitting mentioned here is not justified empirically by showing the generalization error of the model. What does overfitting mean in this context ?  Is the set used for CV very small that it is no longer determining overfitting? Can you show that MV finds the best model compared to CV when the generalization error is evaluated on sufficiently large dataset?  Can you show overfitting in feature space?

(2) It is unclear how useful the method will be as the datasets used are small scale (UCI, MNIST, and CIFAR) and the models used are simple. Moreover the paper describes in Figure 8 that with large scale datasets MV no longer shows similar trends as shown in Figure 3. This makes it difficult to evaluate the significance of this work.

(3) In Table 4 what is the variance in the test accuracy across the different recommended depths?  What does the stability here translate to empirically in terms of model performance? Consider asking similar questions to (1) here.

(4) The paper mentions that the given method is motivated from a software engineering algorithm principle - mutation testing and metamorphic testing. I found this motivation hard to connect. Metamorphic testing is a more definitive test to find bugs in a program (if the relation is violated implies there is a bug) whereas the relation/method proposed by the paper is used in a more relative context (comparing it with nearest hyper-parameters) rather than being a definitive violation of a relation.

(5) How do the two terms - (a) difference in training accuracy of models trained on original data and training accuracy of models trained on mutated data and (b) Accuracy of model trained on mutated data based on the original training data change - change as MV changes empirically?

(6) How is the hit rate in section 4.1 calculated?



**Summary Of The Paper:**

The paper introduces a new method for model validation called Mutation Validation (MV). Briefly, MV mutates training data labels and trains the model on both original and mutated training data. It then uses both these models to get a score that is a function of (1) difference in training accuracy of models trained on original data and training accuracy of models trained on mutated data and (2) Accuracy of model trained on mutated data based on the original training data. The paper explains the larger the score the better the learner fits the training data and hence can be used for model validation.

The paper provides an empirical study to show the effectiveness of MV by comparing it with Cross Validation (CV). In this study they explore 8 different learning algorithms, 18 datasets and 5 types of hyper-parameter tuning tasks.


**Summary Of The Review:**

The idea presented in the paper novel and interesting to me. My major concerns are with the experimental sections - I did not find it convincing enough and I am unsure of its use case given the current results.

---

> ### Author Response · Authors · 2021-11-17
> **Response to Reviewer 11Jy**
>
> > The claim of overfitting in Figure 3 is not justified empirically by showing the generalization error of the model. What does overfitting mean in this context?
>
> **Response**: We adopt the definition of overfitting from the work of Hawkins [1] (with over 2,000 citations) as well as from the wikipedia page of overfitting [2], which is based on Occam’s Razor. In particular, “overfitting is the use of models or procedures that violate parsimony -- that is, that include more terms than are necessary or use more complicated approaches than are necessary” [1]. Overfitting may cause poor generalisation error, but not always [1]. In Figure 3, when MV decreases but CV/test accuracy remains the same, it is a warning to developers that the corresponding depths are larger than necessary for the training data, affecting the interpretability, also making the model vulnerable to training label attacks.
>
> **Revision**: Thanks for your comment. We have made this clear in the second paragraph of the introduction in our revision.
>
> > It is unclear how useful the method will be as the datasets used are small scale (UCI, MNIST, and CIFAR) and the models used are simple.
>
> **Response**: We chose to use datasets that are diverse in size, number of features and number of classes. Small datasets such as UCI datasets are particularly important  to demonstrate the ability of MV in providing warnings for over-complex models. For deep learning datasetes, we use MNIST, fashion-MNIST, CIFAR-10, and CIFAR-100, which are larger than several well-known related studies of deep learning capacity and generalization, such as the paper of Frankle and Carbin (ICLR 2019 best paper) and Arpit et al. (ICML 2017), in which they use *only* MNIST and CIFAR-10.
>
> We agree that not using the largest dataset available (due to equipment limit), such as ImageNet, is a limitation of the current paper. As an initial study on the performance of MV, however, we can and will perform follow-on studies. We have also made all information available to support others should they wish to replicate or investigate further themselves. We look forward to deeper analysis of MV in future from the machine learning community.
>
> **Revision**: We added more explanation about the choice of our datasets in Section 3.
>
> > The paper describes in Figure 8 that with large scale datasets MV no longer shows similar trends as shown in Figure 3.
>
> **Response**: Yes, MV is *expected* to have different patterns on datasets that are much larger than required. The extra training data improve the robustness of the learner to training label noise. The pattern that MV no longer changes when model complexity increases is also a signal to developers that the training data is perhaps larger than necessary.
>
> **Revision**: We added more discussion in Appendix A.3.
>
> > In Table 4 What does the stability here translate to empirically in terms of model performance?
>
> **Response**: We use stability to refer to the fact that out-of-sample accuracy, including test accuracy, can have a large variance across different runs. The large variance has been empirically explored [4], and is one motivation for us to study MV as we mentioned in the introduction.
>
> > Metamorphic testing is a more definitive test to find bugs in a program whereas the relation/method proposed by the paper is used in a more relative context rather than being a definitive violation of a relation.
>
> **Response**: Indeed, we adapted the concept of metamorphic relation in the scenario of model validation, extending it to a qualitative measurement, rather than a simple binary judgement. We believe this adaption is more applicable in machine learning. In addition, at the end of Appendix A.1, we discussed how the usage of MV can be regarded as a metamorphic relation violation problem.
>
> **Revision**: We added more discussion in Appendix A.1.
>
> > How do the two terms - (a) difference in training accuracy of models and (b) Accuracy of model trained on mutated data based on the original training data change - change as MV changes empirically?
>
> **Response**: Following the question, we collected the experimental results in Figure 2 and Figure 3 to analyse their relationship.
>
> **Revision**: The results are added in Appendix A.8. Overall these two terms are observed to have linear relationship with MV.
>
> > How is the hit rate in section 4.1 calculated?
>
> **Response**: The hit rate is the ratio of model recommendations that match the ground truth model.
>
> **Revision**: We have clarified this in Section 4.1.
>
>
> *References*:
>
> [1] Hawkins, Douglas M. "The problem of overfitting." Journal of chemical information and computer sciences 44.1 (2004): 1-12
>
> [2] https://en.wikipedia.org/wiki/Overfitting
>
> [3] Pham, Hung Viet, et al. "Problems and opportunities in training deep learning software systems: an analysis of variance." Proceedings of the 35th IEEE/ACM International Conference on Automated Software Engineering. 2020.

---

> > ### Comment · Reviewer_11Jy · 2021-11-28
> > **Thank you for your response**
> >
> > I have read and thank the authors for their response and intend to keep my score.

---

### Author Response · Authors · 2021-11-19
**General response to all the reviewers**

We thank all the reviewers for their very helpful and valuable feedback. We replied to each reviewer respectively and made an overall revision of our paper addressing EACH point the reviewers have raised. In particular, we have made the following primary changes:

1) We have added model selection results with UCI datasets as an extension of RQ1 in Figure 9.
2) We have compared random label replacement with label swapping in Figure 10.
3) We have added Figure 11 to illustrate how MV gives a warning to developers when the model is more complex than necessary, as well as how it complements test accuracy to find the best parameters that obey the rule of Occam’s Razor (i.e., to find the simpler model among many that have the same generalization error).
4) We have explored the empirical relationship between MV, the new training accuracy, and accuracy changes. We presented the results in Figure 12.
5) We have gone through the whole paper and clarified the vague descriptions that we could find out

Please find more details of our response to each reviewer as well as the full description of our changes below.

---

### Author Response · Authors · 2021-11-30
**Thanks again to all the reviewers and AC**

Thank you very much again for your constructive reviews, your replies, and your time!

---

### Decision · Program_Chairs · 2022-01-20

**Decision:**

Reject

**Comment:**

This paper proposes a model selection technique for classification problems, called mutated validation (MV), based on randomizing the labels of the training set. The idea is interesting but not well served by the presentation of the objectives and the experimental results provided in the paper. The discourse is confusing because it is not clear what is the purpose of model selection here. Usually, model selection aims at finding a model with lowest generalization error. This is a well-defined goal, and the performance regarding this goal can be measured by the expected test error that is usually estimated from a test set. In this paper, the goal of model selection is not precisely defined, but it cannot be as usual, since some models that minimize test error are said to overfit. Most experimental results show that models selected by MV differ from the ones selected by cross-validation (CV), without providing an objective measure of the relative merits of MV and CV (see details below). Although the authors do not make this clear in the paper, they argue in the discussion that the merit of their approach is to select a "simple" model that avoids overfitting. Their message should be clearly stated in the paper, and it should be supported by experiments displaying simplicity *and* test error, or by any experimental result showing the objective benefits of the proposed MV, possibly combined with CV as some of the reviewers suggested. I therefore recommend rejection, but with encouragement to pursue this work, by defining precisely the formal objectives pursued in this work with model selection, and by measuring the benefits of MV on these formalized objectives.

Details:
The arguments in Figure 2 and Table 2 are not substantiated and are clearly not applicable to a model selection mechanism that would aim to select the model with the lowest expected error (or complexity). The "obviously ill-fitting rectangle-shaped decision boundaries" are perfect with respect to expected test error and simply described in Occam's razor terms.
Similarly, most results are subjective:
 - Figure 3 shows that MV and CV are different (on the left), and that CV is a better estimate of the test error (on the right).
 - Figure 4 shows some differences between CV and MV, with no clear way to judge which would be better.
 - Table 4: variance says nothing about relevance; choosing an arbitrary value before seeing the data gives a zero variance.
 - Figure 6, 7: again, subjective result
 - Figure 8: no comparison
 - Figure 10: label swapping or random label replacement is the same in binary classification, there is just a difference in the parameterization (swapping 20% of the labels is the same as randomizing 40% of the labels).
 - Figure 12: I am not sure what is drawn here, but it seems to be related to the training error only.
 -
Regarding objective test results:
 - Figure 5 is an experimental result that shows objective differences between CV and MV, and appear to be marginally in favor of MV for selecting the model with lowest test error (in 3 out of 8 graphs). However, there is no joint optimization on the 2 hyper-parameters: there is not a single data set for which the best value of the test error is identical in the two graphs (wrt dropout, wrt learning rate). In other words, the graphs for dropout rate and/or learning rate are provided for a suboptimal choice of the other hyperparameter.
- Figure 9: CV is more closely related to the test error than MV.
- Figure 11: MV chooses models that do not achieve the highest test accuracy.

As a side note, I think that the proposition could be also positioned with respect to papers that presented similar ideas, where model selection is based on stability, either with unlabeled examples [1] or by modifying the training set [2].

[1] Dale Schuurmans, Finnegan Southey: Metric-Based Methods for Adaptive Model Selection and Regularization. Mach. Learn. 48(1-3): 51-84 (2002)

[2] Olivier Bousquet, André Elisseeff: Stability and Generalization. J. Mach. Learn. Res. 2: 499-526 (2002)